# A common bacterial metabolite elicits prion-based bypass of glucose repression

**David M Garcia[1][†], David Dietrich[2][†], Jon Clardy[2]*, Daniel F Jarosz[1,3]***

[1]Department of Chemical and Systems Biology, Stanford University School of Medicine, Stanford, United States; [2]Department of Biological Chemistry and Molecular Pharmacology, Harvard Medical School, Boston, United States; [3]Department of Developmental Biology, Stanford University School of Medicine, Stanford, United States

**Abstract** Robust preference for fermentative glucose metabolism has motivated domestication of the budding yeast *Saccharomyces cerevisiae*. This program can be circumvented by a protein-based genetic element, the [*GAR*+] prion, permitting simultaneous metabolism of glucose and other carbon sources. Diverse bacteria can elicit yeast cells to acquire [*GAR*+], although the molecular details of this interaction remain unknown. Here we identify the common bacterial metabolite lactic acid as a strong [*GAR*+] inducer. Transient exposure to lactic acid caused yeast cells to heritably circumvent glucose repression. This trait had the defining genetic properties of [*GAR*+], and did not require utilization of lactic acid as a carbon source. Lactic acid also induced [*GAR*+]-like epigenetic states in fungi that diverged from *S. cerevisiae* ~200 million years ago, and in which glucose repression evolved independently. To our knowledge, this is the first study to uncover a bacterial metabolite with the capacity to potently induce a prion.

**\*For correspondence:**
jon_clardy@hms.harvard.edu (JC);
jarosz@stanford.edu (DFJ)

[†]These authors contributed equally to this work

## Introduction

Small molecules commonly drive productive and destructive relationships between species. The breadth of these molecular messages is vividly illustrated by examples ranging from bacterial control of fungal pathogenesis to programming of multi-cellular development (*Alegado et al., 2012*; *Ehrhardt et al., 1992*; *Hogan and Kolter, 2002*; *Hogan et al., 2004*). Small molecule interactions between mammals and their associated gut microbiota may have important consequences for human health (*David et al., 2014*; *Lozupone et al., 2012*). Cross-kingdom interactions also have commercial relevance – for example those that occur between yeast and bacteria in the fermentation of alcoholic beverages (*Bisson et al., 2007*; *Bokulich et al., 2012*; *Boulton et al., 1996*). Pasteur and others since have noted that lactic acid bacteria are a common contaminant in incomplete fermentations (*Boulton et al., 1996*; *Pasteur, 1873*). Indeed, brewers and vintners have long appreciated that some bacteria have the power to both reduce the ethanol content and spoil the taste profile of fermented beverages.

The budding yeast *S. cerevisiae* is a metabolic specialist: it strongly favors glucose as a carbon source even when other carbohydrates are present (*Johnston, 1999*). This is achieved through stringent regulation of enzymes involved in glycolysis and respiration at both the transcriptional and post-transcriptional levels (*Johnston, 1999*; *Zampar et al., 2013*). Such glucose-associated repression of metabolism is a defining feature of *S. cerevisiae*, and many other microorganisms (*Gorke and Stulke, 2008*). A similar preference for glycolytic metabolism, even in the presence of oxygen, is a defining feature of cancer cells, known as the 'Warburg effect' (*Hsu and Sabatini, 2008*; *Liberti and Locasale, 2016*; *Vander Heiden et al., 2009*; *Warburg, 1956*).

**eLife digest** We communicate with each other using speech, writing and physical gestures. But how do bacteria, yeast and other single-celled microbes communicate? In 2014, researchers reported a new example of communication between bacteria and yeast in which the bacteria send a chemical message that has a very long-lasting effect on how the yeast grow in certain environments. This in turn also affected the ability of the bacteria to survive in these environments. The identity of the chemical message produced by the bacteria, however, was not known.

Garcia, Dietrich *et al.* – including one of the researchers from the previous study – used biochemical and genetic approaches to identify the chemical message. The experiments show that the message is a molecule called lactic acid, which is very common in nature and is produced by many bacteria. Garcia, Dietrich *et al.* found out how much lactic acid is needed to alter the growth of brewer's yeast, and which genes in yeast are involved in responding to the message from the bacteria. Further experiments suggest that the ability of yeast and bacteria to communicate using lactic acid is likely to have existed for hundreds of millions of years.

The next step following this work will be to identify other chemical messages used by microbes. The human body is packed with billions of bacterial cells, and in some cases yeast can also take up residence. A future challenge will be to find out if bacteria and yeast inside the human body are able to communicate with each other in ways that could affect our health.

When yeast cultures switch from metabolizing glucose to utilizing a respirative carbon source such as glycerol, they experience a pronounced lag phase known as the diauxic shift. This pause in growth occurs as a new transcriptional program is engaged to favor respiration (*Galdieri et al., 2010*; *Zampar et al., 2013*). The robust nature of this glucose-associated repression can be readily observed by using glucose mimetics. Even in trace quantities, these molecules provide a stable glucose signal, but cannot be metabolized. For example, just 0.05% of the mimetic glucosamine prevents yeast cells from utilizing non-fermentable carbon sources present at 40-fold higher concentrations (*Ball et al., 1976*; *Brown and Lindquist, 2009*; *Kunz and Ball, 1977*).

We recently reported that yeast strains have the ability to overcome such glucose-associated repression of metabolism at frequencies that depend upon the ecological niche from which they were isolated (*Jarosz et al., 2014a*, *2014b*). This allows them to switch from being metabolic specialists to metabolic generalists, gaining the ability to metabolize a wide variety of carbon sources in the presence of glucose. This heritable change in metabolic program is not accompanied by a change in the yeast genotype. Rather, the trait is cytoplasmically transduced from mother cells to their daughters through the inheritance of an altered protein conformation – a prion known as [*GAR*⁺] (named for its ability to reverse glucose associated repression) (*Brown and Lindquist, 2009*). Although the molecular details of [*GAR*⁺] are incompletely understood, several operational tests to identify and modify it are known. [*GAR*⁺]'s nomenclature derives from defining characteristics of prion biology: dominance in genetic crosses (denoted by capital letters) and transmission to all progeny of meiosis (denoted by brackets). The ability to acquire [*GAR*⁺] has been widely conserved among wild *S. cerevisiae* strains and is also present in fungal species separated by at least two hundred million years of evolution (*Jarosz et al., 2014b*).

Under monoculture conditions in the laboratory, acquisition of [*GAR*⁺] occurs much more frequently than spontaneous mutations that reverse glucose repression (*Brown and Lindquist, 2009*). In nature, the precise rates of [*GAR*⁺] appearance correlate with environmental niche from which yeast are derived, suggestive of a bet-hedging mechanism for adaptation in environments with fluctuating carbon stores (*Griswold and Masel, 2009*; *Jarosz et al., 2014b*; *Lancaster and Masel, 2009*). Perhaps most remarkably, [*GAR*⁺] can be induced by cross-kingdom communication with many bacterial species, and this interaction has been conserved over long evolutionary timescales (*Jarosz et al., 2014a*, *2014b*). Notably, this relationship substantiates a model in which [*GAR*⁺]'s benefit as a reversible bet-hedging element would alone be sufficient to motivate its evolutionary retention (*Jarosz et al., 2014b*).

Induction of [$GAR^+$] by bacteria occurs with an extremely high efficiency – orders of magnitude greater than the spontaneous acquisition rate – and the consequences are mutually beneficial. [$GAR^+$] yeast cells gain the ability to metabolize a wider range of carbohydrates, improved uptake of limiting nutrients, and increased chronological lifespan (*Jarosz et al., 2014a*). [$GAR^+$] yeast cells also produce less ethanol, which itself has anti-bacterial properties. This generates a more hospitable environment for the inducing bacteria and allows them to flourish in fermentations where they would otherwise have perished (*Jarosz et al., 2014a*); bacterial species that strongly induce [$GAR^+$] are common contaminants in failed wine fermentations (*Bisson et al., 2007*; *Boulton et al., 1996*; *Jarosz et al., 2014a*). When yeast and inducing bacteria are co-cultured on solid medium, induction of [$GAR^+$] occurs along a spatial gradient emanating from the bacteria. Conditioned medium generated from cultures of these bacteria also induces the prion (*Jarosz et al., 2014a*). These results and others have led us to propose that inducing bacteria produce a diffusible signal that promotes [$GAR^+$] in yeast.

Here we report that the common bacterial metabolite lactic acid can serve as this inducing signal. Different concentrations of lactic acid elicit distinct and heritable [$GAR^+$]-like phenotypes that have the properties of 'strong' and 'weak' variants of the prion. Lactic acid also has the capacity to induce [$GAR^+$]-like epigenetic states in evolutionarily distant organisms in which glucose repression has evolved independently. Our results suggest that [$GAR^+$] induction may provide a molecular explanation for Pasteur's observations, and indicate that links between lactate and diversification of metabolic strategies have been repeatedly employed to modulate the collective behavior of eukaryotic cells. More importantly, we for the first time uncover a bacterially secreted molecule that induces a prion. We do so using a straightforward approach with few technical prerequisites, which we propose could be used for discovering other natural molecules that potentiate epigenetic transformations.

## Results

### Lactic acid induces robust and heritable reversal of glucose repression

Diverse bacteria have the capacity to induce [$GAR^+$] in neighboring yeast cells through a cross-kingdom chemical communication. To characterize the diffusible element that was responsible for eliciting this change in yeast metabolism, we employed activity guided fractionation. We first grew large-volume cultures of the inducing bacteria *S. gallinarum* and *L. innocua* to saturation in rich media containing glycerol as a principle carbon source (YP-glycerol). After removing the bacteria from the media by filtration to create a sterile conditioned medium, we assessed its capacity to induce growth on media containing glycerol (2%) and glucosamine (0.05%), hereafter abbreviated as GLY + GlcN. GlcN provides a stable, non-metabolizable glucose signal (*Ball et al., 1976*; *Kunz and Ball, 1977*). Serial dilutions of naïve [$gar^-$] yeast did not grow on GLY + GlcN alone over a five day period, but pre-treatment with the conditioned medium caused the yeast cells to grow robustly.

We then purified the extracts from each inducing bacterium using reversed-phase high performance liquid chromatography (HPLC) on a C18 column. We collected fractions and monitored their ability to induce yeast cells to grow in GLY + GlcN. Pre-treatment with this purified extract afforded growth to a similar degree as *S. gallinarum* itself (*Figure 1A*). After these two rounds of purification, the inducing activity eluted in a single fraction as a homogenous peak. We combined active fractions from multiple injections and analyzed this peak by NMR and mass spectrometry. Both analyses revealed that the active fraction contained high concentrations of lactic acid (*Figure 1B*; *Figure 1—figure supplement 1*).

Lactic acid, a small organic molecule, exists in two distinct forms: L-(+)-lactic acid (L-LA) and its mirror image D-(–)-lactic acid (D-LA). Both occur naturally. Some bacteria, including the class known as lactic acid bacteria (LAB), have the ability to produce one or both forms from carbohydrates (*Eiteman and Ramalingam, 2015*; *Salminen et al., 2004*). Eukaryotes typically can metabolize both isomers, sometimes to varying degrees (*Adeva-Andany et al., 2014*; *Pajot and Claisse, 1974*). Malolactic fermentation by LAB in winemaking can typically yield up to 2.5 g/L of lactic acid, or 0.25% (*Boulton et al., 1996*); in other fermented foods, concentrations can exceed 10 g/L, or >1% (*Fleming et al., 1988*; *Plengvidhya et al., 2007*). Moreover, many bacterial species that we previously found to strongly induce [$GAR^+$] are known LAB (*Jarosz et al., 2014a*).

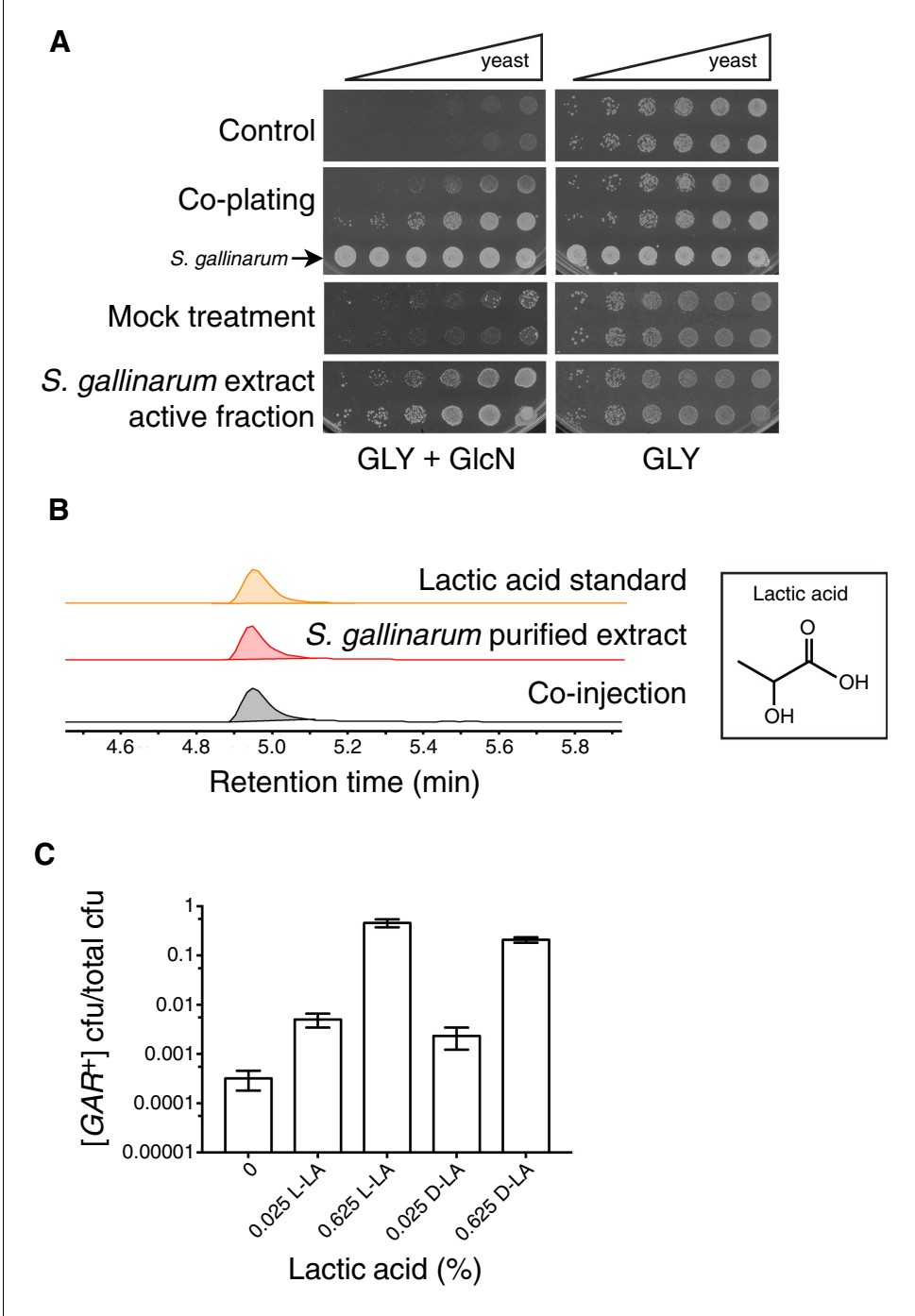

**Figure 1.** Identification of the small molecule that induces [*GAR*+] in a concentration dependent manner. (**A**) *S. gallinarum* spotted next to yeast on GLY + GlcN (YP with 2% glycerol and 0.05% glucosamine) induced growth of naïve *S. cerevisiae* in a spatially dependent manner. An extract prepared from a *S. gallinarum* conditioned medium induced [*GAR*+] with similar strength. Each panel shows two biological replicates of yeast colonies spotted in five-fold serial dilutions from saturated cultures. (Bacterial spots are undiluted.) (**B**) Multiple-reaction monitoring-mass spectrometry (*m/z* 89.0200, 43.1000) traces of pure lactic acid (top), the purified active fraction from *S. gallinarum* conditioned medium (middle) and a co-injection of active fraction spiked with pure lactic acid (bottom). Results indicate that lactic acid is present in the active fraction (see Materials and methods). Note that y-axis scale differs between the top two traces due to the difference of lactic acid concentration in the pure standard, but they are normalized here for ease of viewing. (**C**) Plating assay (see Materials and methods) showing that both L- and D-isomers of lactic acid strongly induce [*GAR*+] in a concentration dependent manner. Plotted are the fraction of

*Figure 1 continued on next page*

*Figure 1 continued*

total colony forming units (CFUs) that grew on GLY + GlcN (mean with standard deviation from three biological replicates). Ten-fold dilutions were used for plates lacking lactic acid; 1000-fold and 10,000-fold dilutions were used for plates containing 0.025% LA and 0.625% LA, respectively.

The following figure supplements are available for figure 1:

**Figure supplement 1.** Flow chart depicting the activity guided fractionation strategy used to identify the small molecule that induces [*GAR*+].

**Figure supplement 2.** Growth curves—of cells that start as [*gar*−] at the outset of the experiment—in synthetic complete media (Sigma-Aldrich) containing glucosamine, with or without 0.1% L-lactic acid added.

**Figure supplement 3.** Growth rates of [*GAR*+] and [*gar*−] cells do not differ in the presence of lactic acid.

To establish whether lactic acid was sufficient to induce heritable reversal of glucose repression in *S. cerevisiae*, we tested the ability of yeast to grow on GLY + GlcN agar plates supplemented with concentrations of L- or D-lactic acid at 0.025 or 0.625%. We saw robust reversal of glucose repression in this range from both stereoisomers, in a concentration dependent manner (*Figure 1C*). D-lactic acid induced cells to acquire the phenotype slightly more rapidly (data not shown). Addition of 0.05% lactic acid did not increase growth on YP alone, eliminating the possibility that the yeast were simply metabolizing the lactic acid (data not shown). Induction was also robust in synthetic complete medium, further establishing that it was due to the added lactic acid, rather than unknown components of the medium (e.g. yeast extract, peptone, or an unknown metabolite therein; *Figure 1—figure supplement 2*). Finally, we addressed the possibility that the presence of lactic acid could select for pre-existing [*GAR*+] cells, as opposed to true induction of [*gar*−] cells into the [*GAR*+] state. We examined the growth rates of isogenic [*GAR*+] cells and [*gar*−] cells in liquid YP-glycerol (GLY) alone or in YP-glycerol supplemented with 0.35% lactic acid. We did not observe any significant prion-dependent differences in growth (*Figure 1—figure supplement 3*), supporting an induction mechanism rather than selection of a pre-existing population.

## Reversal of glucose-associated repression is due to [*GAR*+]

Glucose repression can be commonly circumvented in *S. cerevisiae* by induction of the [*GAR*+] prion, but it can less frequently arise from genetic mutations (*Ball et al., 1976*; *Kunz and Ball, 1977*) and, in principle, from other heritable epigenetic modifications (*Jarosz et al., 2014a*, *2014b*). We thus tested whether the reversal of glucose repression that we observed in response to lactic acid was due to [*GAR*+]. To do so we examined hallmarks of prion-based inheritance: dominance in genetic crosses and the ability to be heritably reduced or eliminated by perturbations in chaperone activity (*Shorter and Lindquist, 2005*; *Wickner et al., 2004*). We first mated yeast cells that had been induced with 0.1% lactic acid to naïve isogenic mating partners and selected for diploids. All diploids we tested grew on GLY + GlcN medium, and moreover exhibited the same unusual pattern of semi-dominance (*Figure 2A*) – in which the diploid cells exhibit a strength of reversal of glucose repression that is intermediate between the two haploid parents – that is highly characteristic of both spontaneous and bacterially-induced [*GAR*+] (*Brown and Lindquist, 2009*; *Jarosz et al., 2014a*). We next sporulated diploid cells induced with lactic acid, dissected multiple tetrads, and examined the ability of the resulting spores to grow on GLY + GlcN. Nearly all spores we tested (29 of 31) exhibited the [*GAR*+] phenotype, in agreement with the non-Mendelian, dominant inheritance previously noted for both spontaneous and bacterially induced [*GAR*+] cells (*Figure 2—figure supplement 1*).

[*GAR*+] is unusual among known prions in that it is not influenced by perturbation of the Hsp104 disaggregase (*Brown and Lindquist, 2009*; *Jarosz et al., 2014a*). Instead it is sensitive to perturbation of Hsp70 function (*Brown and Lindquist, 2009*; *Jarosz et al., 2014a*, *2014b*). We assembled multiple colonies that had been induced with lactic acid to grow on GLY + GlcN plates. We then transformed them with a plasmid that constitutively expressed a dominant negative Hsp70 variant (Ssa1-K69M, or Hsp70$^{DN}$). We passaged these cells five times on media that selected for the

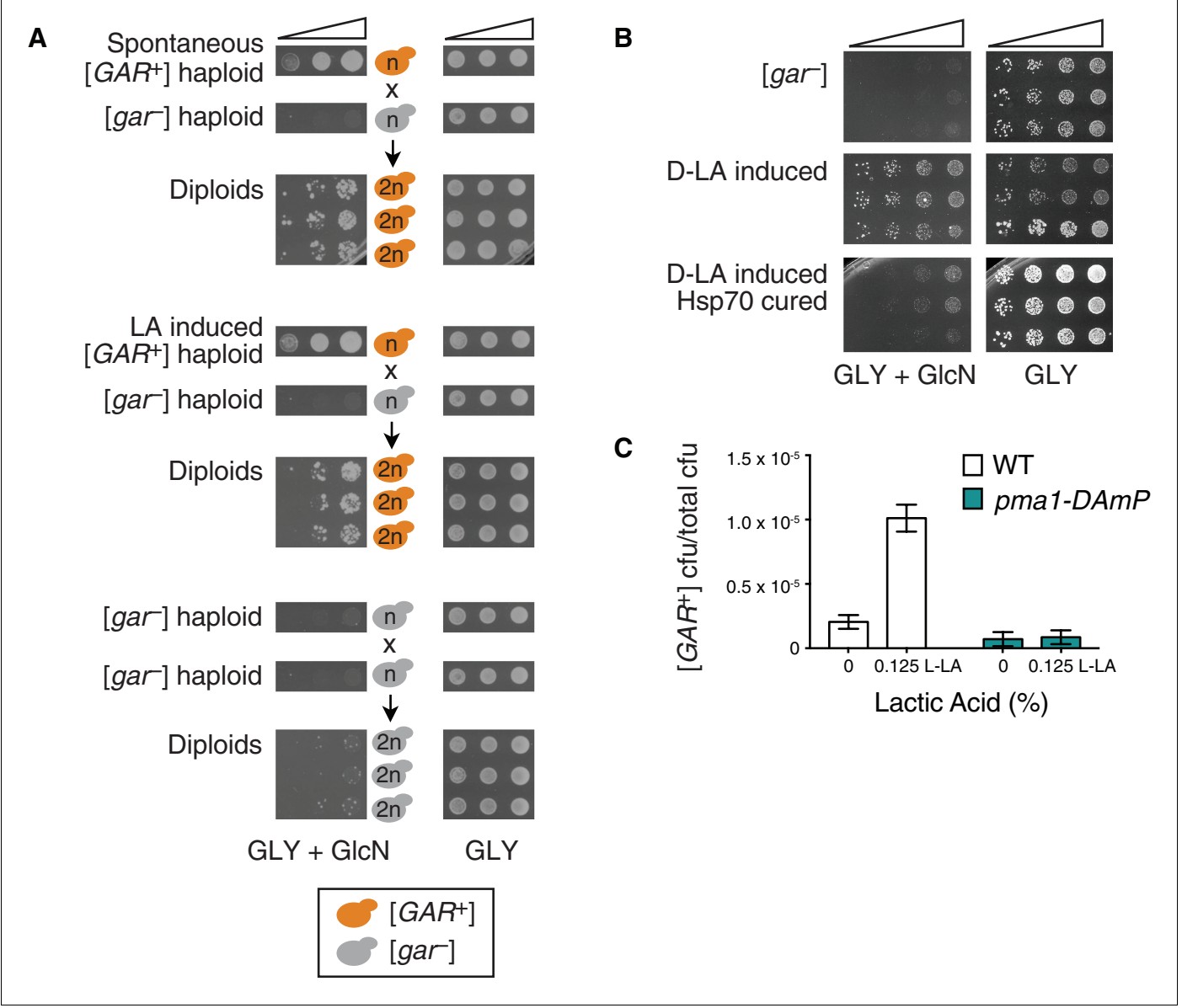

**Figure 2.** Lactic acid induced [*GAR*+] has the same prion-like features as spontaneous [*GAR*+]. (**A**) Dominance in crosses between [*GAR*+] strains and [*gar*−] strains is seen in colonies isolated from a GLY + GlcN plate (spontaneous [*GAR*+]), as well as those induced with 0.1% D-lactic acid (LA-induced [*GAR*+]). Five-fold serial dilutions are shown for three biological replicates; orange coloring indicates strains exhibiting the [*GAR*+] phenotype. (**B**) Lactic acid-induced cells can be cured of the [*GAR*+] phenotype with transient expression of a dominant negative Hsp70 variant (Ssa1-K69M or Hsp70[DN]). Single colonies were propagated alone or with a plasmid expressing Hsp70[DN] for ~125 generations, bottlenecked each ~25 generations. After verifying plasmid loss, colonies were again tested for their capacity to grow on GLY + GlcN medium. Each image shows five-fold serial dilutions for three biological replicates. (**C**) A strain with diminished expression of *PMA1*, *pma1-DAmP*, has reduced spontaneous [*GAR*+] acquisition and is not induced by lactic acid. Lactic acid-induction frequencies differ from those in *Figure 1C* due to strain background BY4741, which has a lower frequency of [*GAR*+] acquisition (*Brown and Lindquist, 2009*). Plotted as in *Figure 1C*.

The following figure supplements are available for figure 2:

**Figure supplement 1.** Both spontaneous and lactic acid-induced [*GAR*+] diploids transmit the phenotype through meiosis to nearly all spores, demonstrating non-Mendelian inheritance.

**Figure supplement 2.** Both spontaneous [*GAR*+] cells and lactic acid-induced [*GAR*+] *S. cerevisiae* cells exhibited sensitivity to the Hsp70 inhibitor myricetin (50 µM) on GLY + GlcN medium.

plasmid. We then passaged them three times on non-selective plates (and confirmed plasmid loss) before retesting them on GLY + GlcN plates. This regimen cured multiple independent clones of their induced [*GAR*⁺] phenotype, indicating that the lactic acid induced colonies have the same Hsp70-dependency as spontaneous and bacterially induced [*GAR*⁺] cells (*Figure 2B*). We also grew induced cells on plates containing an Hsp70 inhibitor, myricetin (*Chang et al., 2011*; *Koren et al., 2010*), at concentrations that did not affect growth on GLY medium. This regimen, which has been shown to inhibit the growth of both spontaneous and bacterially induced [*GAR*⁺] on GLY + GlcN (*Jarosz et al., 2014b*), also reduced the ability of these colonies to circumvent glucose repression that had been induced by lactic acid (*Figure 2—figure supplement 2*).

We next tested whether our lactic acid induced isolates exhibited genetic characteristics related to proteins originally linked to spontaneous [*GAR*⁺]. *PMA1* encodes the essential major plasma membrane proton pump in yeast; in [*GAR*⁺] cells Pma1 adopts an altered conformation and associates more strongly with another protein, Std1 (*Brown and Lindquist, 2009*). Because *PMA1* is essential, we employed *pma1-DAmP*, a degron allele that reduces the amount of protein expressed (*Breslow et al., 2008*). Cells harboring this allele had a low frequency of spontaneous prion acquisition, and could not be induced to acquire [*GAR*⁺] by exposure to lactic acid (*Figure 2C*). Thus, the heritable bypass of glucose repression induced by lactic acid has the same prion-like patterns of inheritance and depends upon the same proteins as spontaneous and bacterially induced [*GAR*⁺]. Although this lactic acid-induced epigenetic state could in principle differ from the spontaneous prion in other unknown ways, for simplicity we hereafter refer to it as [*GAR*⁺].

## Influence of yeast lactic acid metabolism on [*GAR*⁺] induction

Lactic acid can be imported into yeast via saturable ("active") and unsaturable ('passive') mechanisms (*Casal et al., 1999*; *Pacheco et al., 2012*; *Paiva et al., 2013*). Both depend upon the transcription factor *SOK2*, and the active mechanism is ultimately mediated by the monocarboxylate transporter *JEN1*. Jen1 is required for yeast to grow on lactic acid as a sole carbon source, but is down-regulated during glucose repression (*Casal et al., 1999*; *Pacheco et al., 2012*), and its transcript was not abundant in [*GAR*⁺] cells compared to control cells (*Brown and Lindquist, 2009*; *Jarosz et al., 2014a*). Deletion of *JEN1* greatly reduces growth on glycerol compared to wild-type cells, precluding its emergence in the previous screens for genes affecting [*GAR*⁺] (*Brown and Lindquist, 2009*; *Jarosz et al., 2014a*). We therefore tested if lactic acid could induce *jen1Δ* cells to acquire [*GAR*⁺]. Deletion of *JEN1* did not influence [*GAR*⁺] induction by lactic acid at either low or high concentrations (*Figure 3A,B*), indicating that active transport is not required. We also tested whether enzymes reported to be involved in lactic acid metabolism in the *Saccharomyces* Genome Database (*Cherry et al., 2012*) could influence the ability of this molecule to induce [*GAR*⁺]. Individual deletion of these various lactic acid dehydrogenases and transhydrogenases (*DLD1, DLD2, DLD3*, and *CYB2*) also had no significant effect on [*GAR*⁺] induction by lactic acid (*Figure 3A,B*). Thus, the cellular mechanisms engaged by lactic acid to induce [*GAR*⁺] do not require its active import or its metabolism.

*SOK2* previously emerged as a hit in the screen for yeast mutants that were impaired in their ability to acquire [*GAR*⁺] upon induction by the bacterium *Staphylococcus hominis* (*Jarosz et al., 2014a*). We replicated this result using *S. gallinarum* in a co-plating assay on GLY + GlcN (*Figure 4A*). We then tested whether loss of *SOK2* impaired the capacity of lactic acid alone to induce [*GAR*⁺] in yeast cells. It did: yeast cells in which *SOK2* was deleted were not induced by lactic acid (*Figure 4B*). Thus, the previously described passive import of lactic acid (*Paiva et al., 2013*) is likely the dominant means through which this inducing signal is transmitted from bacteria to neighboring fungi.

## Lactic acid concentration influences the strength of the [*GAR*⁺] activity state

Since environmental conditions can exert a strong influence on the amount of lactic acid produced by bacteria, we next tested differing amounts of lactic acid, over a larger concentration regime. Even the lowest concentration we tested in this experiment, 0.015%, increased the number of colonies that arose on a GLY + GlcN plate by nearly three orders of magnitude, such that approximately 5% of cells grew into medium-sized colonies (*Figure 5A*). Increasing concentrations of lactic acid up

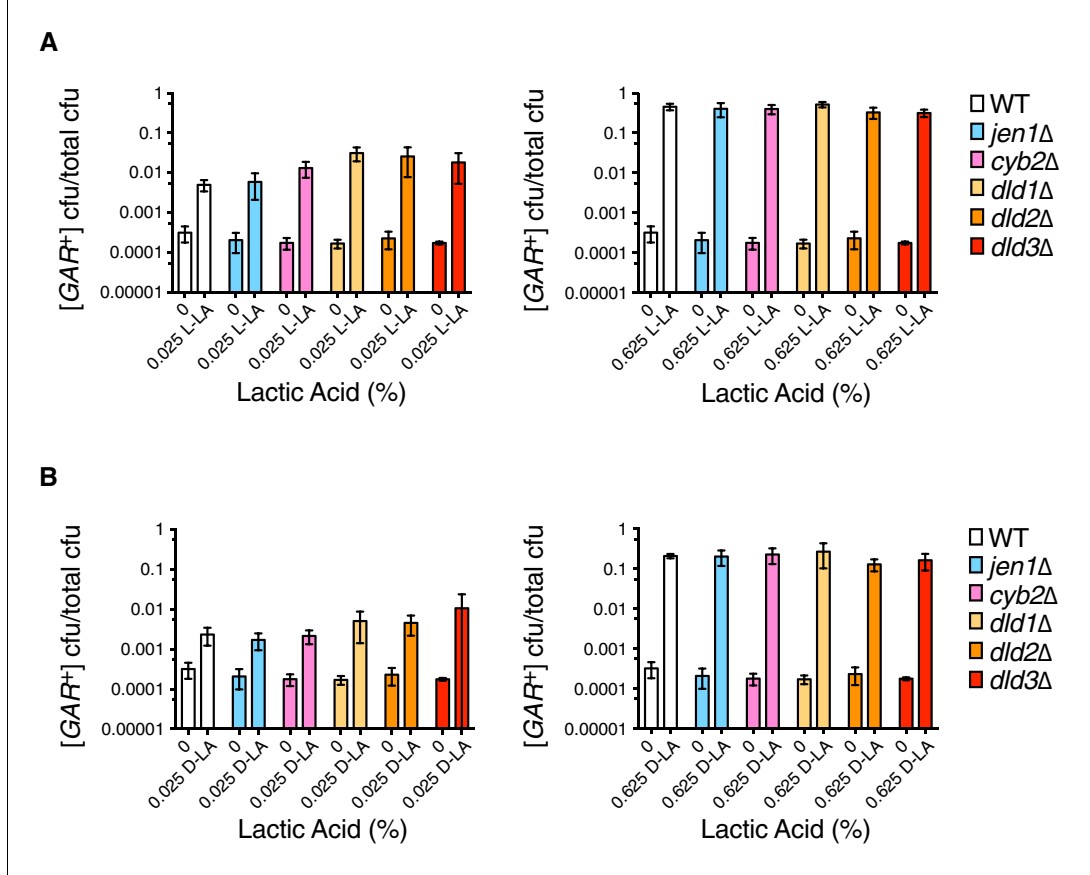

**Figure 3.** Analysis of induction frequencies in deletion mutants of genes with roles in lactic acid metabolism. (**A**) Plating assay (see Materials and methods) measuring induction of [$GAR^+$] at a low (left) or high concentration (right) of L-lactic acid. Strains are in the W303 background, and cell dilutions match those in *Figure 1C*. We note that annotations of *DLD2* and *DLD3* as D-lactate dehydrogenases have recently been challenged by data suggesting they are instead transhydrogenases (*Becker-Kettern et al., 2016*). Only *DLD1* is required for yeast to grow on D-lactic acid as a sole carbon source. Plotted the same as in *Figure 1C*. B.) Plating assay measuring induction of [$GAR^+$] by D-lactic acid. Otherwise as in (**A**).

to 0.25% did not strongly increase the degree of induction. However, 1% lactic acid caused an even stronger induction, such that nearly all cells in the population grew into large colonies on GLY + GlcN media (*Figure 5A*).

Like virtually all prions, [$GAR^+$] can exist in multiple 'strains' – different heritable and stable activity states. When they arise spontaneously on selective GLY + GlcN plates, 'strong' [$GAR^+$] colonies are larger than 'weak' [$GAR^+$] colonies (*Brown and Lindquist, 2009*; *Jarosz et al., 2014a*). Both activity states are heritable over long biological timescales (*Brown and Lindquist, 2009*; *Jarosz et al., 2014a*). We picked small-sized colonies that had been induced with low concentrations of lactic acid, and large colonies that had been induced with higher concentrations of lactic acid and propagated them for ~75 generations on non-selective rich medium (YPD). We then returned them to GLY + GlcN plates. The small colonies that had originally been induced with low concentrations of lactic acid grew modestly on GLY + GlcN, in a manner that mimicked 'weak' [$GAR^+$] strains (*Figure 5B*). The large colonies that had originally been induced with high concentrations of lactic acid grew robustly on GLY + GlcN, in a manner that mimicked 'strong' [$GAR^+$] strains (*Figure 5B*). Thus, alterations in the concentration of lactic acid produced by inducing bacteria may engage different heritable activity states, or so-called 'strains', of the [$GAR^+$] prion.

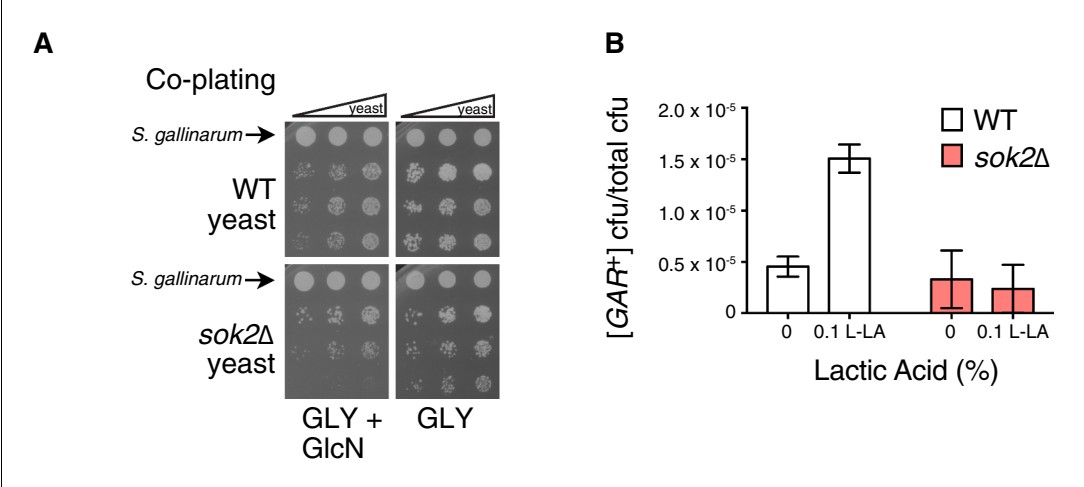

**Figure 4.** Full induction by *S. gallinarum* or lactic acid requires the yeast transcription factor *SOK2*. (**A**) *S. gallinarum* spotted next to either wild type or *sok2Δ* yeast cells on GLY + GlcN, or control medium GLY. (**B**) Plating assay (see Materials and methods) comparing spontaneous and lactic acid-induced [*GAR*+] frequency in wild type and *sok2Δ* cells. Lactic acid-induction frequencies differ from those in *Figure 1C* due to strain background BY4741, which has a lower frequency of [*GAR*+] acquisition (*Brown and Lindquist, 2009*). Plotted the same as in *Figure 1C*.

## Lactic acid induces [*GAR*+]-like epigenetic states in evolutionarily distant fungal species

Glucose repression has evolved multiple times in the fungal lineage (*Hagman and Piskur, 2015*). *Dekkera bruxellensis*, an organism used in the production of lambic beers that diverged from *S. cerevisiae* 200 million years ago, uses a distinct mechanism for engaging stringent glucose repression (*Rozpedowska et al., 2011*; *Woolfit et al., 2007*). We previously established that despite this evolutionary distance, *D. bruxellensis* can spontaneously acquire a [*GAR*+]-like state to become a metabolic generalist (*Jarosz et al., 2014b*). This [*GAR*+]-like state can also be induced by cross-kingdom chemical communication with bacteria. Although the molecular lexicon of cross-kingdom communication can evolve extraordinarily rapidly, the simplicity and ubiquity of lactic acid led us to investigate whether it might also influence heritable reversal of stringent glucose repression in *D. bruxellensis*.

We exposed naïve *D. bruxellensis* strains to lactic acid, guided by our results from *S. cerevisiae*. Concentrations well within the physiological range of lactic acid produced by bacteria (0.02%) induced robust growth on GLY + GlcN (*Figure 6A*). We picked colonies from these selections and propagated them for ~75 generations on non-selective rich medium (YPD). In each case, the trait was heritable: induced colonies maintained the ability to circumvent glucose repression (*Figure 6B*).

The [*GAR*+]-like states that arise spontaneously or are induced by bacteria in *D. bruxellensis* depend strongly on the activity of Hsp70 chaperones (*Jarosz et al., 2014b*). We therefore tested whether the *D. bruxellensis* colonies induced by lactic acid to circumvent glucose repression were sensitive to Hsp70 inhibition. We plated serial dilutions of *D. bruxellensis* cells that had been induced with lactic acid on GLY + GlcN plates and on GLY + GlcN plates containing the Hsp70 inhibitor myricetin (*Chang et al., 2011*; *Koren et al., 2010*). We used a concentration of this inhibitor that did not significantly affect growth on GLY alone. This treatment significantly reduced the ability of these induced colonies to circumvent glucose repression (*Figure 6B*), just as it does for the [*GAR*+]-like states that arise spontaneously in this organism (*Jarosz et al., 2014b*). Thus, across a vast evolutionary distance, bacterial production of lactic acid can engage [*GAR*+]-like mechanisms to heritably convert neighboring fungi from metabolic specialists to metabolic generalists.

## Discussion

More than a century ago, Louis Pasteur documented what vintners have long since appreciated: the presence of lactic acid producing bacteria in fermentations can often lead to their failure

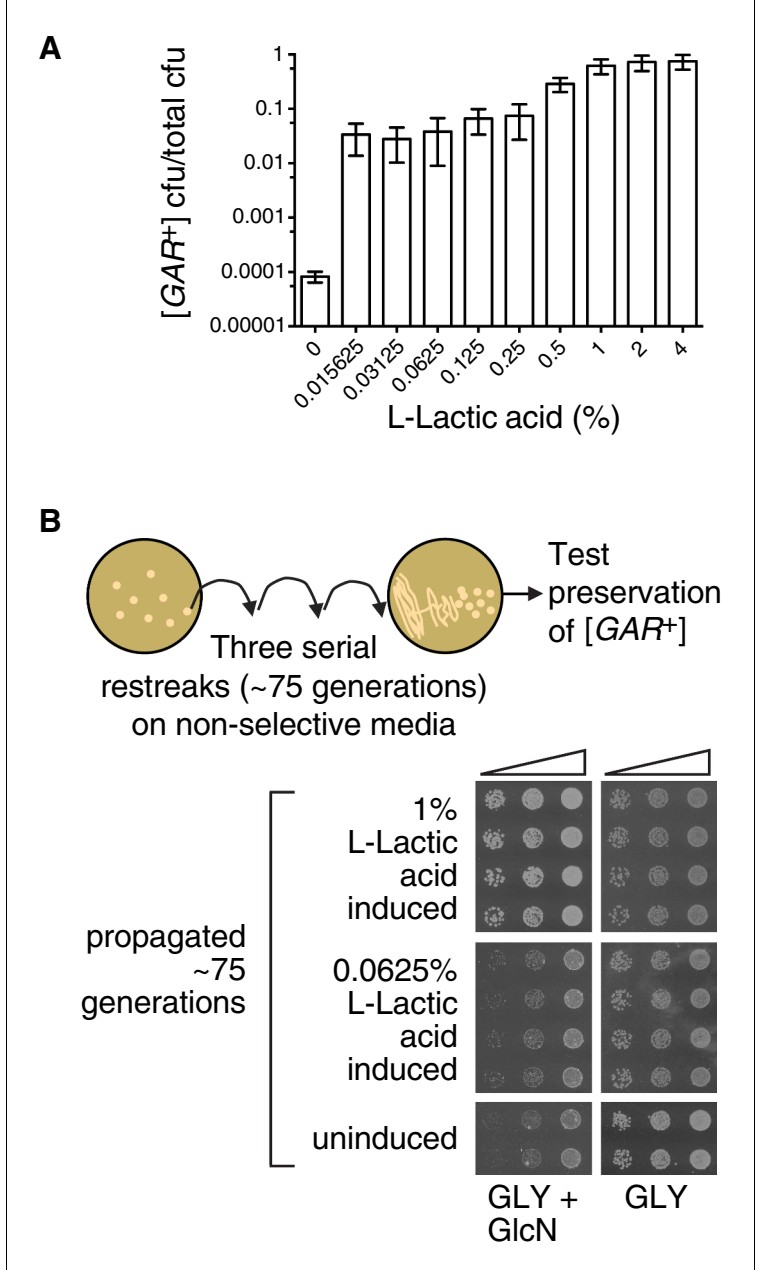

**Figure 5.** The strength of the [*GAR*⁺] phenotype depends on the concentration of lactic acid used for induction. (**A**) Plating assay (see Materials and methods) showing that L-lactic acid strongly induced [*GAR*⁺] over a wide concentration range. Plotted the same as in *Figure 1C*. Ten-fold dilutions were used for plates lacking lactic acid and 10,000-fold dilutions were used for plates containing lactic acid (**B**) Single [*GAR*⁺] colonies were picked from two representative concentrations in (A), propagated (restreaked) on non-selective medium (GLY) for approximately 75 generations, and then re-spotted on GLY + GlcN in five-fold serial dilutions. Four biological replicates are shown for each induced group, and two for uninduced cells. Note the growth in the third spots (highest dilution) of 0.0625% L-LA induced cells on GLY + GlcN.

(*Pasteur, 1873*). Our observations suggest that the ability of lactic acid to elicit the [*GAR*⁺] prion could provide a molecular mechanism for those observations. The [*GAR*⁺] prion is found in evolutionarily distant fungal species, and can confer strong selective advantage – converting metabolic specialists to metabolic generalists in a manner that improves fitness in complex microbial communities

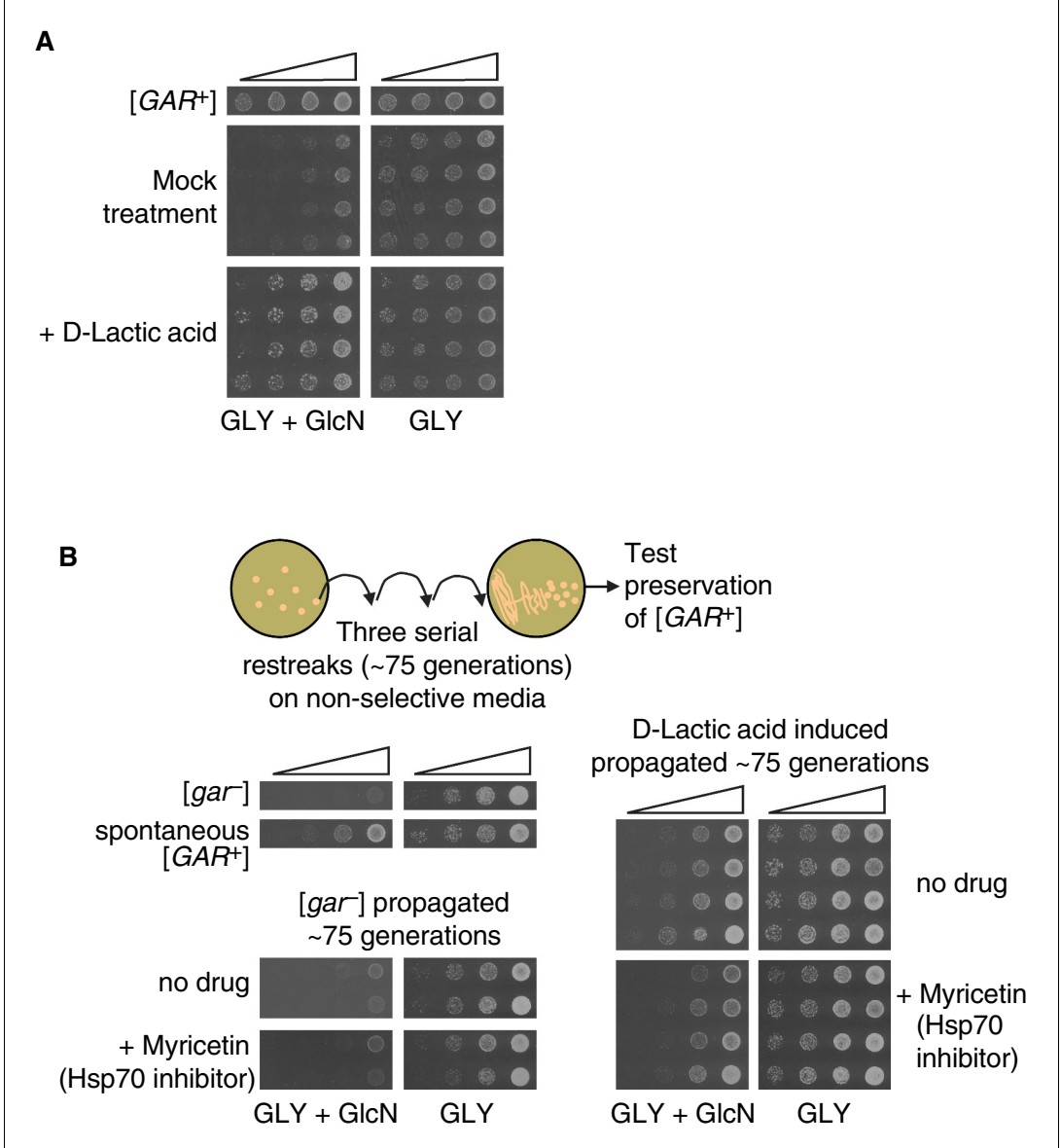

**Figure 6.** Induction, maintenance, and chaperone dependence of a lactic acid-induced [GAR⁺]-like state is conserved in the evolutionary distant yeast *Dekkera bruxellensis*. (**A**) Induction of a [GAR⁺]-like state in *D. bruxellensis*. [gar⁻] cells were plated on GLY + GlcN (containing 0.1–0.15% glucosamine) medium with or without 0.02% D-lactic acid. Control cells harbor spontaneous [GAR⁺]. For each of the treatment images, four biological replicates are shown, in five-fold serial dilutions. (**B**) Maintenance and chaperone dependence of the lactic acid-induced [GAR⁺]-like state in *D. bruxellensis*. Cells induced in A. were propagated non-selectively for ~75 generations by restreaking on YPD plates, then subsequently retested for maintenance of the prion phenotype on GLY + GlcN (0.1–0.15% glucosamine), with or without the Hsp70 inhibitor myricetin (50 µM). Four biological replicates are shown for each induced group, and two shown for the uninduced and propagated [gar⁻] cells.

(*Jarosz et al., 2014a*, *2014b*). Our data link the common bacterial metabolite lactic acid, both structurally simple and widespread in nature, to this ancient switch in metabolic strategies.

Recent evidence points to important signaling roles for lactic acid in human cancer cells, in the context of the Warburg effect (*Adeva-Andany et al., 2014*; *Dhup et al., 2012*; *Fiaschi et al., 2012*; *Liberti and Locasale, 2016*). Other studies have linked LAB to benefits for human health and disease prevention, however it remains unclear how much these effects are due to the effects of lactic acid on host cells (*Kim et al., 2007*; *Masood et al., 2011*; *Tiptiri-Kourpeti et al., 2016*). In at least

some cases, LAB appear to exert their health effects by influencing other microbes, for example, modulating the virulence of *Clostridium difficile* (*Lee et al., 2013*; *Yun et al., 2014*).

Many mysteries remain regarding the genetic and biochemical features that lead to the [*GAR*+] prion; findings have so far been limited to its initial characterization (*Brown and Lindquist, 2009*) and discovery of its regulation by environmental cues (*Jarosz et al., 2014a*). Our discovery of a connection between passive lactic acid uptake and the ability to induce [*GAR*+] (via our results with *SOK2*) provides new insight into the initiation of this complex cross-kingdom chemical communication. Further molecular understanding of induction by lactic acid, and its links to the unusual biochemistry that fuels [*GAR*+], stands as a goal for future studies.

Several lines of evidence suggest that other molecules may modulate [*GAR*+] induction by lactic acid. Although we were able to fully reconstitute induction with lactic acid alone, the kinetics were slightly faster in conditioned media. Moreover, some of the bacteria that induce [*GAR*+] in *S. cerevisiae* do not induce the [*GAR*+]-like state in *D. bruxellensis*, even though high levels of lactic acid can induce the prion in each of these organisms. An intriguing but plausible explanation is that other bacterially secreted factors influence perception of the lactic acid signal, and that the effects of these molecules may also be species specific. Indeed, there is precedent for this behavior in choanoflagellates, where two bacterially secreted lipids were required to recapitulate full induction of rosette, multicellular colony formation (*Alegado et al., 2012*; *Woznica et al., 2016*). Finally, although we have shown that *S. gallinarium* induces [*GAR*+] via lactic acid, it is entirely possible that other molecules could also fuel this cross-kingdom interaction in diverse ecological niches. There is also precedent in mechanisms of quorum sensing for competition among inducers secreted by different bacteria (*Drescher et al., 2014*; *Even-Tov et al., 2016*).

Perhaps most remarkably, our study raises the possibility that other prion-like epigenetic states could be regulated by common metabolites, at times in service of cross-species communication. Prior studies have highlighted the fact that general stressors that perturb protein homeostasis can modestly elevate the frequency of cells harboring the [*PSI*+] or [*MOT3*+] prions (*Holmes et al., 2013*; *Tyedmers et al., 2008*). These intrinsic links between stress responses and prion switching ensure that subpopulations of cells can diversify their phenotypes when they are ill-suited to the environment. Our findings extend this biology to a more concerted, switch-like behavior in which prions can serve as a target for rapid and effective cross-species communication.

Biological niches like the human body harbor microbial communities that are far more complex than those we have studied here (*Lozupone et al., 2012*). It is striking that the robust connection we have discovered between a simple metabolite and a heritable change in metabolic strategy has been missed despite a multitude studies in this model organism. Our observations thus underscore the potential of cross-kingdom molecular communication to illuminate biological processes, in particular its capacity to elicit heritable, epigenetic transformations of biological traits. Recent progress in metabolomics and metagenomics motivate deeper exploration of such interactions between species, at both the molecular and systems levels.

## Materials and methods

### Yeast and bacterial media and strain maintenance

Bacteria strains were cultured on LB agar (Research Products International, Mt. Prospect, IL and BD-Difco, Franklin Lakes, NJ) plates. Yeast strains were cultured on either: (1) YPD agar or liquid (RPI and BD-Difco), (2) YP-Glycerol agar (1% Yeast Extract (BD Bacto), 2% Peptone (BD BBL/Difco), 2% glycerol (Sigma-Aldrich, St. Louis, MO and Amresco, Solon, OH), 2% agar (IBI Scientific, Peosta, IA]), or (3) "'GLY + GlcN'' agar (same mixture as YP-Glycerol) which is supplemented with 0.05% glucosamine hydrochloride (Sigma-Aldrich) (0.1–0.15% for *D. bruxellensis*) that was added from a filter-sterilized 5% stock once media cooled to ~60°C after autoclaving. We noticed that the stringency of selection for [*GAR*+] cells on GLY + GlcN media was dependent on the source of yeast extract and peptone, and therefore recommend the aforementioned suppliers, as other sources for unknown reasons sometimes imposed much weaker selection. Myricetin (Sigma-Aldrich) was made into a stock solution of 15.7 mM in 100% ethanol, and then diluted to a final concentration of 50 µM in agar medium once cooled to ~60°C after autoclaving.

Strains were maintained using standard microbiological techniques. Bacteria and *S. cerevisiae* strains were grown up in 2–5 mL cultures at 30°C shaking or rotating overnight for most experiments; for experiments requiring smaller volumes they were grown in 200 μL volumes in microtiter plates. All *D. bruxellensis* strains were grown at room temperature until saturation for liquid or sufficiently large colonies for agar plates, typically requiring at least 5 days of growth.

Serial dilutions, starting from saturated cultures, were performed in 96-well plates using multichannel pipettes, and cells were spotted gently onto agar plates using a blot replicating tool (V&P Scientific VP404A, San Diego, CA). For *S. cerevisiae* strains, GLY plates were grown for 2 days before imaging. GLY + GlcN plates were grown 5–7 days before imaging. *D. bruxellensis* strains required several extra days of growth for single colonies to become visible.

## Activity guided fractionation

Bacterial strains that induced yeast growth on GLY + GlcN media were grown in large scale to purify the active component. Strains were grown from single colonies in 5 mL LB cultures overnight at 30°C and with 250 rpm shaking. The inoculum was then transferred to Erlenmeyer flasks containing 200 mL YP-Glycerol and incubated for 72 hr at 250 rpm, 30°C. The media were centrifuged (30 min at 6000 rpm, Beckman (Brea, CA) JLA 8.1 rotor), and passed through a 0.22 μm filter. The filter-sterilized media was acidified to pH ~1 using 2M HCl, and extracted with a threefold volumetric excess of ethyl acetate. The organic extracts were dried over sodium sulfate, concentrated on a rotary evaporator, suspended in 2–4 mL water, and neutralized with sodium hydroxide. The extract was purified by preparative HPLC using a gradient of acetonitrile (solvent B, Fisher Scientific (Waltham, MA), containing 0.1% formic acid) in milli-Q water (solvent A, containing 0.1% formic acid) on a reversed-phase C18 column (Phenomenex (Torrance, CA) Luna 5U, 100A, 250 × 21.29 mm). The following program was used: 10 mL/min; ramp from 0 to 20 min of 2% to 35% B, followed by a 10 min wash cycle; collect 0.5 min fractions from 7 to 13 min.

Collected fractions were dried in a speed-vac, suspended in water (~5 mL) and passed through a 0.22 μm filter. Active fractions were identified by assaying their ability to induce growth of YDJ1034 in typically non-permissive GLY + GlcN media containing crude bacterial extracts or HPLC fractions (10 to 25% of total media volume). Fractions that supported yeast growth in GLY + GlcN were combined and subjected to subsequent rounds of semi-preparative HPLC using a gradient of acetonitrile (containing 0.1% formic acid – solvent B) in water (containing 0.1% formic acid – solvent A) on a reversed-phase C18 column (Phenomenex Luna 5U, 100A, 250 x 10.00 mm). The following program was used: 2.4 mL/min; ramp from 0 to 20 min of 2% to 10% B, followed by a 15 min wash cycle; collect 0.2 min fractions from 8.4 to 9.4 min. Fractions were analyzed by high-resolution HPLC-MRM-mass spectrometry and NMR spectroscopy. The active fractions contained high levels of lactic acid, which was confirmed by NMR and by co-injection with a purified standard (*Figure 1B*).

## Prion induction using lactic acid

Multiple sources of lactic acid yielded similar results. We tested sodium D-lactate (Sigma-Aldrich 71716), sodium L-lactate (Sigma-Aldrich L7022), D-lactic acid zinc salt (MP Biomedicals, Santa Ana, CA), and L-(+)-lactic acid (Sigma-Aldrich L6402). Excess NaCl was tested in control reactions to mimic the sodium chloride produced during naturalization of sodium lactate and found to produce no phenotypic differences. Lactic acid was added to GLY + GlcN medium at indicated concentrations before autoclaving, and pH was balanced if necessary to ensure it was within ~0.1 units or less of control media lacking lactic acid (pH ~6.8).

For the plating assay, *S. cerevisiae* W303 strains were plated with 100 μL volumes for all plates. For BY4741 strains, 250 μL volumes were plated for GLY + GlcN plates +/– lactic acid (no cell dilution), while 100 μL was used for GLY plates. Cell dilution was empirically determined to obtain plates containing between approximately 25 and 300 colonies per plate, which were counted using an aCOLyte colony counter (Synbiosis, Cambridge, England). For all GLY plates, a 100,000-fold dilution of saturated culture was plated. For GLY + GlcN plates for W303 strains, a 10-fold dilution of saturated culture was plated; for GLY + GlcN plates containing lactic acid, dilutions of 1000-fold or 10,000-fold were plated to achieve a suitable colony number for W303 strains.

Induction frequency was calculated by normalizing the number of colony forming units (CFUs) on the GLY + GlcN plates to the total possible number on GLY plates after accounting for cell dilutions,

reported as [$GAR^+$] cfu/total cfu. We noted some variability in values of [$GAR^+$] cfu/total cfu from experiment to experiment (e.g. for spontaneous rate between *Figure 1C* and *Figure 5A*), however the qualitative relationship between lactic acid concentration and increased [$GAR^+$] cfu/total cfu was maintained across all dilutions tested. For induction of *D. bruxellensis* strains, cells were spotted in serial dilutions on plates containing lactic acid. Single colonies were picked from the highest dilution and propagated as described in figure legends.

### Strain constructions

Diploids were constructed by crossing indicated BY4741 haploids to the BY4742 parental strain (ATCC, Manassas, VA) by mixing a bead of cells of each strain together on a YPD plate and growing overnight. A small globule of this cell mixture was then restreaked to single colonies on SD-Lys-Met agar plates to select for diploids.

Deletion mutants in the W303 background were created by amplifying a *KanMX* construct from corresponding gene deletion mutants from the BY4741 library, using primers listed in *Supplementary file 1C*. Approximately 500 ng of purified construct was transformed into YDJ1034, and integrants were selected on agar plates containing G418 (Fisher Scientific). All gene deletions were confirmed by PCR using primers listed in *Supplementary file 1C*.

### Prion curing and sporulations

Curing of the [$GAR^+$] phenotype (*Figure 2B*) was achieved by transformation of lactic acid induced cells with PDJ281 (Hygromycin resistant, *Supplementary file 1D*), restreaking multiple transformants to single colonies on YPD plates containing 400 ug/mL hygromycin (Gold Biotechnology, Olivette, MO) a total of five times, then restreaking in the same manner on YPD plates twice before confirming each clone had lost the plasmid by testing on YPD+hygromycin. These clones were grown in liquid YPD cultures before testing on GLY + GlcN media.

Spontaneous and lactic acid-induced [$GAR^+$] diploids (W303 background, *Figure 2—figure supplement 1*) were isolated from single colonies grown on GLY + GlcN or GLY + GlcN + lactic acid plates and grown in Pre-SPO liquid media (YPD with 6% glucose) for 3 days at room temperature. Cells were then pelleted and washed twice in SPO media (1% Potassium Acetate (Sigma-Aldrich), 320 mg CSM-Met powder (Sunrise Science, San Diego, CA), 20 mg Methionine (Sigma-Aldrich) per liter), and then diluted ten-fold to obtain 3 mL cultures in SPO media. These cultures were incubated on a rotary wheel for one week at room temperature, before dissecting tetrads on a Singer Instruments MSM400 (Somerset, UK). Recovered spores were grown to saturation in liquid YPD before testing on GLY + GlcN media.

### Data collection and display

Yeast plates were imaged using an Epson Perfection V700 Photo Scanner (Long Beach, CA). Plots were made using PRISM 6 software (GraphPad, La Jolla, CA).

## Acknowledgements

We want to acknowledge Soledad Larios (Stanford), Linda Bisson (UC Davis), and members of the Clardy and Jarosz laboratories for resources, discussions, and feedback on the manuscript. This work was supported by a grant from the NIH (R01 GM086258) to JC, and by an NSF-CAREER award (NSF-MCB116762) and Searle Scholars Award (14-SSP-210) to DFJ. DMG was supported by post-doctoral fellowships from the Ford Foundation and the NIH (F32-GM109680), as well as the Burroughs Wellcome Fund Postdoctoral Enrichment Program.

## Additional information

#### Competing interests

JC: Reviewing editor, *eLife*. The other authors declare that no competing interests exist.

## Funding

| Funder | Grant reference number | Author |
| --- | --- | --- |
| National Institutes of Health | R01 GM086258 | Jon Clardy |
| National Science Foundation | CAREER award, NSF-MCB116762 | Daniel F Jarosz |
| Searle Scholars Program | Searle Scholars Award, 14-SSP-210 | Daniel F Jarosz |
| National Institutes of Health | F32 GM109680 | David M Garcia |
| Ford Foundation | Postdoctoral Fellowship | David M Garcia |
| Burroughs Wellcome Fund | Postdoctoral Enrichment Program | David M Garcia |

The funders had no role in study design, data collection and interpretation, or the decision to submit the work for publication.

## Author contributions

DMG, DD, DFJ, Conception and design, Acquisition of data, Analysis and interpretation of data, Drafting or revising the article; JC, Conception and design, Analysis and interpretation of data, Drafting or revising the article

## Author ORCIDs

David M Garcia, http://orcid.org/0000-0003-0600-9527
David Dietrich, http://orcid.org/0000-0002-6477-4078
Jon Clardy, http://orcid.org/0000-0003-0213-8356
Daniel F Jarosz, http://orcid.org/0000-0003-3497-5888

# Additional files

## Supplementary files

• Supplementary file 1. (A) Bacterial strains used in this study. (B) Yeast strains used in this study. (C) Primers used in this study. (D) Plasmids used in this study.

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
