## [Decision Letter]

Thank you for submitting your article "A common bacterial metabolite elicits prion-based bypass of glucose repression" for consideration by *eLife*. Your article has been favorably evaluated by Jessica Tyler (Senior Editor) and three reviewers, one of whom is a member of our Board of Reviewing Editors. The reviewers have opted to remain anonymous.

The reviewers have discussed the reviews with one another and the Reviewing Editor has drafted this decision to help you prepare a revised submission.

Summary:

This succinct paper reports the solution to a mystery posed by the authors' earlier work on induction of the [*GAR*^+^] 'prion' by a low molecular weight molecule. That the inducer is lactic acid makes sense given the known ability of lactic acid-producing bacteria to induce [*GAR*^+^] as well as the historical context vis-à-vis Pasteur's observations. The data that demonstrate that lactic acid is a [*GAR*^+^] inducer are well controlled and nicely presented although the 'induction vs. selection' argument that is always raised in such studies should ideally have been explicitly addressed in the experimental design and resulting Discussion. The manuscript is clearly written and experimental design is straightforward.

However, several issues need to be addressed before this manuscript can be accepted for publication.

Essential revisions:

1) Authors need to show that the glucosamine resistant colonies induced by lactic acid indeed contain a [*GAR*^+^] prion. The evidence provided in the manuscript includes a dominance in genetic crosses and inhibition by myricetin, an inhibitor of Hsp70. However, this is insufficient to conclude that the observed phenotype is controlled by a prion. First, authors need to show that in the lactic acid induced colonies, this phenotype shows non-Mendelian inheritance in meioisis and/or is transmitted by cytoplasmic infection (cytoduction).

2) Second, they need to show that myrecitin irreversibly cures [*GAR*^+^] rather than inhibiting it. it isn't clear from the paper if the phenotypic inhibition by inhibition of HSP70 is due to "curing" some cells of the prion, or a general inhibition of phenotype in all cells. Also it isn't clear if this "prion" inhibition is present only during HSP70 inhibition or if transient inhibition of HSP70 has a lasting effect on the so called [*GAR*^+^] phenotype.

3) Third, at least in some isolates obtained from induction by lactic acid, either genetic or biochemical characterization needs to be performed showing that the phenotype under investigation is indeed related to proteins implicated in originally described [*GAR*^+^]. For example, two proteins have been identified as being associated with [*GAR*^+^], one (Pma1 p) is a major plasma membrane proton pump while the other (Std1p) is a transcription factor that among other targets, regulates hexose transporters. Does lactic acid influence the levels/activity of either of these in any plausible way? (In case if non-Mendelian inheritance/cytoduction and chaperone sensitivity are proven, but clear identification of the obtained isolates as [*GAR*^+^] is not done, results might still be publishable, but authors should be more careful in interpretations and should talk about the non-Mendelian prion-like elements similar to previously described [*GAR*^+^], rather than about [*GAR*^+^] prion per se.

4) The same comments apply to [*GAR*^+^] isolates obtained in Dekkera yeast. As detailed characterization of such isolates in Dekkera is more difficult than in Saccharomyces, authors should probably change wording and refer to them more accurately, e. g. as glucosamine resistant colonies similar to [*GAR*^+^].

5) Authors talk about mitotic stabilities of [*GAR*^+^] isolates, but don't show actual data. These data need to be shown.

6) A further complexity alluded to by the authors in their brief Discussion is the possibility that there are other molecular 'co-factors' that are part of the induction mechanism – although clearly lactic acid can induce [*GAR*^+^] without added (but unidentified) 'co-factors'.

7) What is disappointing is that the authors do not really attempt to come up with a plausible model based on their data and what was previously published about [*GAR*^+^] and its induction. The very nature of the [*GAR*^+^] determinant itself remains a mystery; it certainly shows some of the key genetic properties of the other well established yeast prions, yet there are also distinct differences such as a lack of reliance on Hsp104-mediated disaggregation for propagation and the lack of an association with a specific amyloid-forming protein. The authors also mention that [*GAR*^+^] is semi-dominant in [*GAR*^+^] x [*gar*^-^] crosses; do other prions show this behaviour?

[Editors' note: further revisions were requested prior to acceptance, as described below.]

Thank you for resubmitting your work entitled "A common bacterial metabolite elicits prion-based bypass of glucose repression" for further consideration at *eLife*. The manuscript has been improved but there are some remaining issues that need to be addressed before acceptance, as outlined below:

1) The last sentence of the Abstract sounds too strong, as the paper by Holmes et al. 2013 has claimed that the [*MOT3+*] prion can be induced by ethanol. Even though an argument of induction versus selection could possibly be made there (especially as [*MOT3+*] increases ethanol resistance), such a statement requires an elaboration and is more appropriate for the Discussion rather than Abstract. Possibly a substitution of "small molecule" with a "normal bacterial metabolite" (or "a small molecule produced by bacteria") would suffice.

2) This would be helpful to include the growth curves in the presence of lactic acid, shown in Response to reviewers, into a supplement.

---

## [Author Response]

*Summary:*

*This succinct paper reports the solution to a mystery posed by the authors' earlier work on induction of the [GAR^+^] 'prion' by a low molecular weight molecule. That the inducer is lactic acid makes sense given the known ability of lactic acid-producing bacteria to induce [GAR^+^] as well as the historical context vis-à-vis Pasteur's observations. The data that demonstrate that lactic acid is a [GAR^+^] inducer are well controlled and nicely presented although the 'induction vs. selection' argument that is always raised in such studies should ideally have been explicitly addressed in the experimental design and resulting Discussion.* To address the induction vs. selection argument, we have examined whether [*GAR*^+^] cells have accelerated growth rates compared to [*gar*^–^] cells in the presence of inducing concentrations of lactic acid, which might raise the possibility that we could be favoring selection for [*GAR*^+^] colonies in our assays to measure induction frequency.

We grew four biological replicates of [*GAR*^+^] or [*gar*^–^] cells in YP-glycerol or YP-glycerol + 0.35% D-lactic acid liquid media, measuring OD600 every 6 minutes over 4 days using an automated microplate reader. As shown in Figure 1—figure supplement 3, [*GAR*^+^] cells neither grew better than [*gar*^–^] cells in glycerol alone nor when lactic acid was added to the medium. While we cannot exclude the possibility that a [*gar*^–^] cell switching to a [*GAR*^+^] state may introduce some immediate but short-lived growth advantage in lactic acid, we posit that this difference would need to be almost inconceivably large to result in the even the lowest levels of induction that we measured.

*The manuscript is clearly written and experimental design is straightforward.*

*However, several issues need to be addressed before this manuscript can be accepted for publication.*

*Essential revisions:*

*1) Authors need to show that the glucosamine resistant colonies induced by lactic acid indeed contain a [GAR^+^] prion. The evidence provided in the manuscript includes a dominance in genetic crosses and inhibition by myricetin, an inhibitor of Hsp70. However, this is insufficient to conclude that the observed phenotype is controlled by a prion. First, authors need to show that in the lactic acid induced colonies, this phenotype shows non-Mendelian inheritance in meioisis and/or is transmitted by cytoplasmic infection (cytoduction).*

We thank the reviewer for this suggestion – it is a critical point. We have now added data showing non-Mendelian inheritance of the induced phenotype in meiosis. These data appear in a supplemental figure associated with Figure 2 (Figure 2—figure supplement 1).

*2) Second, they need to show that myrecitin irreversibly cures [GAR^+^] rather than inhibiting it. it isn't clear from the paper if the phenotypic inhibition by inhibition of HSP70 is due to "curing" some cells of the prion, or a general inhibition of phenotype in all cells. Also it isn't clear if this "prion" inhibition is present only during HSP70 inhibition or if transient inhibition of HSP70 has a lasting effect on the so called [GAR^+^] phenotype.*

We agree that it is important to test whether transient perturbations in Hsp70 function are sufficient to permanently eliminate the induced [*GAR*^+^] phenotype. To rigorously address this question we have employed a dominant negative Hsp70 variant (Ssa1-K69M) in this revision. We transiently inhibited Hsp70 activity by expressing this dominant negative variant of Hsp70 from a plasmid and then ‘cured’ the cells of this construct, restoring normal Hsp70 function. Just as it did with spontaneous and bacterially-induced [*GAR*^+^] (Jarosz et al., 2014a), this regimen eliminated the [*GAR*^+^] phenotype induced by lactic acid. These data are now provided in Figure 2.

We have moved the myricetin data from *S. cerevisiae* to Figure 2—figure supplement 2, in part to motivate the use of this method for *D. bruxellensis [GAR*^+^]-like cells described in Figure 6. (*D. bruxellensis* is not genetically tractable.) Accordingly, we have altered the language we use to describe the [*GAR*^+^]-like state in this organism.

*3) Third, at least in some isolates obtained from induction by lactic acid, either genetic or biochemical characterization needs to be performed showing that the phenotype under investigation is indeed related to proteins implicated in originally described [GAR^+^]. For example, two proteins have been identified as being associated with [GAR^+^], one (Pma1 p) is a major plasma membrane proton pump while the other (Std1p) is a transcription factor that among other targets, regulates hexose transporters. Does lactic acid influence the levels/activity of either of these in any plausible way? (In case if non-Mendelian inheritance/cytoduction and chaperone sensitivity are proven, but clear identification of the obtained isolates as [GAR^+^] is not done, results might still be publishable, but authors should be more careful in interpretations and should talk about the non-Mendelian prion-like elements similar to previously described [GAR^+^], rather than about [GAR^+^] prion per se.*

Spontaneous [*GAR*^+^] arises from alterations in the structure of Pma1 that influence its association with Std1. We have added data to address the requirement for Pma1 in lactic acid’s capacity to induce the [*GAR*^+^] state, now in Figure 2. *PMA1* is an essential gene, so we took advantage of a strain containing a degron allele that reduces its mRNA levels significantly (Breslow et al., 2008). We show that this strain produces colonies on GLY + GlcN plates at a low frequency that is not further induced by lactic acid. Thus we conclude that the ability of lactic acid to induce the [*GAR*^+^] state depends on the level of Pma1 in the cell, just as spontaneous prion acquisition does.

*4) The same comments apply to [GAR^+^] isolates obtained in Dekkera yeast. As detailed characterization of such isolates in Dekkera is more difficult than in Saccharomyces, authors should probably change wording and refer to them more accurately, e. g. as glucosamine resistant colonies similar to [GAR^+^].*

We have altered the language accordingly when referring to the “[*GAR*^+^]-like” state in *Dekkera*.

*5) Authors talk about mitotic stabilities of [GAR^+^] isolates, but don't show actual data. These data need to be shown.*

Thanks for noticing our wording error. We have edited this section to reflect the intention of our experiments.

*6) A further complexity alluded to by the authors in their brief Discussion is the possibility that there are other molecular 'co-factors' that are part of the induction mechanism – although clearly lactic acid can induce [GAR^+^] without added (but unidentified) 'co-factors'.*

We have expanded on this point in the Discussion.

*7) What is disappointing is that the authors do not really attempt to come up with a plausible model based on their data and what was previously published about [GAR^+^] and its induction. The very nature of the [GAR^+^] determinant itself remains a mystery; it certainly shows some of the key genetic properties of the other well established yeast prions, yet there are also distinct differences such as a lack of reliance on Hsp104-mediated disaggregation for propagation and the lack of an association with a specific amyloid-forming protein. The authors also mention that [GAR^+^] is semi-dominant in [GAR^+^] x [gar^-^] crosses; do other prions show this behaviour?*

We agree there are many mysteries that remain to be uncovered for the [*GAR*^+^] prion. Moreover, surprisingly little is known about how yeast cells respond to lactic acid compared to other carbon sources. This lack of knowledge in both areas hampers our ability to propose a mechanistic model that is more than just conjecture. We now allude to this in the Discussion, highlighting our result that *SOK2* provides a link between the two responses.

The similarities and differences in biochemical properties of [*GAR*^+^] compared to other prions is indeed interesting. Semi-dominance is key among them. This work raises another interesting point of comparison that fascinates us – can other prions be induced by common environmental factors? Of course, we would not expect that lactic acid could induce all yeast prions, but discovery of inducers for other prions could substantially revise our understanding of the ecological dynamics at play in prion biology. We also now incorporate these lines of thinking in the Discussion.

[Editors' note: further revisions were requested prior to acceptance, as described below.]

*Thank you for resubmitting your work entitled "A common bacterial metabolite elicits prion-based bypass of glucose repression" for further consideration at eLife. The manuscript has been improved but there are some remaining issues that need to be addressed before acceptance, as outlined below:*

*1) The last sentence of the Abstract sounds too strong, as the paper by Holmes et al. 2013 has claimed that the [MOT3+] prion can be induced by ethanol. Even though an argument of induction versus selection could possibly be made there (especially as [MOT3+] increases ethanol resistance), such a statement requires an elaboration and is more appropriate for the Discussion rather than Abstract. Possibly a substitution of "small molecule" with a "normal bacterial metabolite" (or "a small molecule produced by bacteria") would suffice.*

We have adjusted the language in the Abstract to reflect this useful point. We have also altered the Discussion slightly to reflect the nuance raised here by the reviewers. Holmes et al. provide as evidence favoring induction that exposure to 12% ethanol “was not overtly toxic to cells, indicating that ethanol did not merely select for pre-existing [*MOT3*^+^] cells,” but they establish that these effects are derived from general perturbations in protein homeostasis. This is consistent with prior work by Tyedmers et al., which found that multiple general proteostasis stressors could induce another prion, [*PSI*^+^]. We now make these points explicitly in the Discussion.

*2) This would be helpful to include the growth curves in the presence of lactic acid, shown in Response to reviewers, into a supplement.*

Thank you for the suggestion. We uploaded this data as Figure 1—figure supplement 3, added the figure legend, and tracked changes in the main text to reflect its inclusion.